# The Contribution of Scientists to the Research in Biosphere Reserves in Slovakia

Jana Špulerová [1,*], Veronika Piscová [2] and Noemi Matušicová [2]

1    Institute of Landscape Ecology of the Slovak Academy of Sciences, Štefánikova 3, 81499 Bratislava, Slovakia
2    Institute of Landscape Ecology of the Slovak Academy of Sciences, Branch Nitra, Akademická 2,
     94901 Nitra, Slovakia
*    Correspondence: jana.spulerova@savba.sk

**Abstract:** This review is aimed at summarizing the current state of knowledge of biosphere reserves (BRs) in Slovakia and assessment of research activities undertaken there and how they contribute to the mission and fulfillment of the goals of the designation process to the World Network of BRs. We based our methodological approach on the literature review of the studies found in the scientific database Web of Science through keyword searches. The 121 studies were characterized by research subject, BR function examined, and contribution to the development of which particular aspect of BR. Most of the studies focused on biodiversity protection, management of BR, land use changes, and scenario modeling. The strengths of BR in Slovakia are a long history and continuity of research, close cooperation with some scientists and institutions, case studies of BR included in international projects, existing examples of participatory studies, and a wide range of research topics. An important contribution to research is that provided by existing long-term monitoring sites. The transboundary BRs in particular are involved in developing international collaborations within the World Network of BRs. We summarized the results of the literature review and gave a scientist's perception of the development of BR in a SWOT analysis, including recommendations for further development in the form of a discussion of opportunities and threats.

**Keywords:** biosphere reserve; research activities; man and biosphere; environmental; ecological; economic; social aspects



## 1. Introduction

The concept of the intergovernmental scientific program Man and Biosphere (MAB), initiated in 1974 by the United Nations Educational, Scientific and Cultural Organization (UNESCO), was to combine research in the field of natural and social sciences to optimize the protection of biodiversity and natural capital of the biosphere [1]. This initiative focused on the designation of biosphere reserves (BR) in protected areas of various bioregions of the world as a coordinated network of permanent study sites for research into issues of biodiversity conservation interlinked with man and sustainable development and associated research, education, and training. It was, therefore, a scientific program promoting BRs as special places for people and nature and not a new category of nature protection. A key component of MAB's objective is to achieve a sustainable balance between the sometimes-conflicting goals of, and the three main functions of BR: conserving biodiversity, promoting economic development, and maintaining associated cultural values. The updated version of the MAB Strategy for 2015–2025 modifies the MAB Programme to fit the new context of the 2030 Agenda for Sustainable Development and its Sustainable Development Goals (SDGs). The strategic objectives for the period 2015–2025 are oriented to (1) conserving biodiversity, restoring and enhancing ecosystem services, and fostering the sustainable use of natural resources; (2) contributing to building sustainable, healthy, and equitable societies, economies and thriving human settlements in harmony with the biosphere; (3) facilitating biodiversity and sustainability science, education for sustainable development and capacity

building; and (4) supporting mitigation and adaptation to climate change and other aspects of global environmental change. This is also reflected in the designation of three zones of biosphere reserves: (1) Core, (2) Buffer, and (3) Transitional Zones. Since the launch of the UNESCO MAB Programme, the number of BRs in the world has increased to 738 (including 20 transboundary BRs) in 134 countries [2]. The gap between the designation objectives of BRs and their management is often very wide [3]. Therefore, the BRs' management needs to be reinforced by focused research and by application of research results for sustainable development and support for mitigation of and adaptation to climate change and other global environmental changes. This support should take the form of appropriate management measures for the favorable conservation status of valuable habitats, e.g., grassland, wetland, or forest, to stop strong deteriorating trends and loss of biodiversity. The Lima Action Plan approved in 2016 also emphasizes the important role of research and international collaborations within the World Network of BR for societal transformation. Research about BR management effectiveness can contribute to a better understanding of the existing gap between the BR concept and its implementation. However, there is limited understanding of where and how research on BRs' management effectiveness has been conducted, what topics are investigated, and what the main findings are [4]. In Slovakia, many studies and monitoring programs have been carried out on the territory of the four Slovak BRs: Slovenský Kras, Poľana, the Tatra Transboundary Biosphere Reserve (Poland/Slovakia) and the East Carpathians Transboundary Biosphere Reserve (Poland/Slovakia/Ukraine). A permanent monitoring of species and habitats of conservation importance is regularly carried out in Slovakia and is evaluated on the national level in accordance with Article 17 of the Habitats Directive every 7 years [5,6]. A combination of older research with newly-published or ongoing studies creates the possibility of evaluating current results and development and changes in the BR in the context of other socio-economic activities.

Our review aimed at summarizing the current state of knowledge on BR research in Slovakia and assessment of research activities undertaken there, based on a systematic literature review of scientific papers indexed by the Web of Science and major topics in BR research. We investigated the content of publications, subjects, main findings of the studies, and how they contribute to the mission and fulfillment of the goals of the designation process and building the World Network of Biosphere Reserves.

## 2. Materials and Methods

### 2.1. Study Area

Four BRs have been designated in Slovakia: Slovenský Kras (1977), Poľana (1990), Tatra Transboundary Biosphere Reserve (Poland/Slovakia) (1992), and East Carpathians Transboundary Biosphere Reserve (Poland/Slovakia/Ukraine) (1992) (Figure 1, Table 1). They are protected according to Act no. 543/2002 Col. on the Protection of Nature and Landscape as territories of international importance, and at the same time as protected areas of national importance—Poľana is a Protected Landscape Area (PLA), and the others are National parks (NP).

#### 2.1.1. Slovensky Kras Biosphere Reserve

Slovenský kras BR is situated in the south of Slovakia, adjacent to the Aggtelek Biosphere Reserve in Hungary. Slovenský kras consists of seven plateaus, ranging between 400 and 900 m above sea level. The Karst area represents a typical central European plateau karst of the temperate climatic zone, with almost all surface and underground karst phenomena (karren fields, dolines, uvalas, blind and semi-blind valleys, canyons, gorges, edge polje, caves, abysses, ponors, karst springs). Most of the BR belongs to temperate warm and moderately wet climatic regions with cold winters. The depression forms climatic and vegetation inversion. More than 1400 caves are known in the Slovak karst. The Domica cave is the most important cave place of the Bukk-Mountain Culture of the Neolithic (4000 years B.C.) in Slovakia. It stretches for 5080 m. In 2002, the level of

nature protection of the majority of the Slovak Karst territory was increased and declared a national park with an area of 34,611 ha.

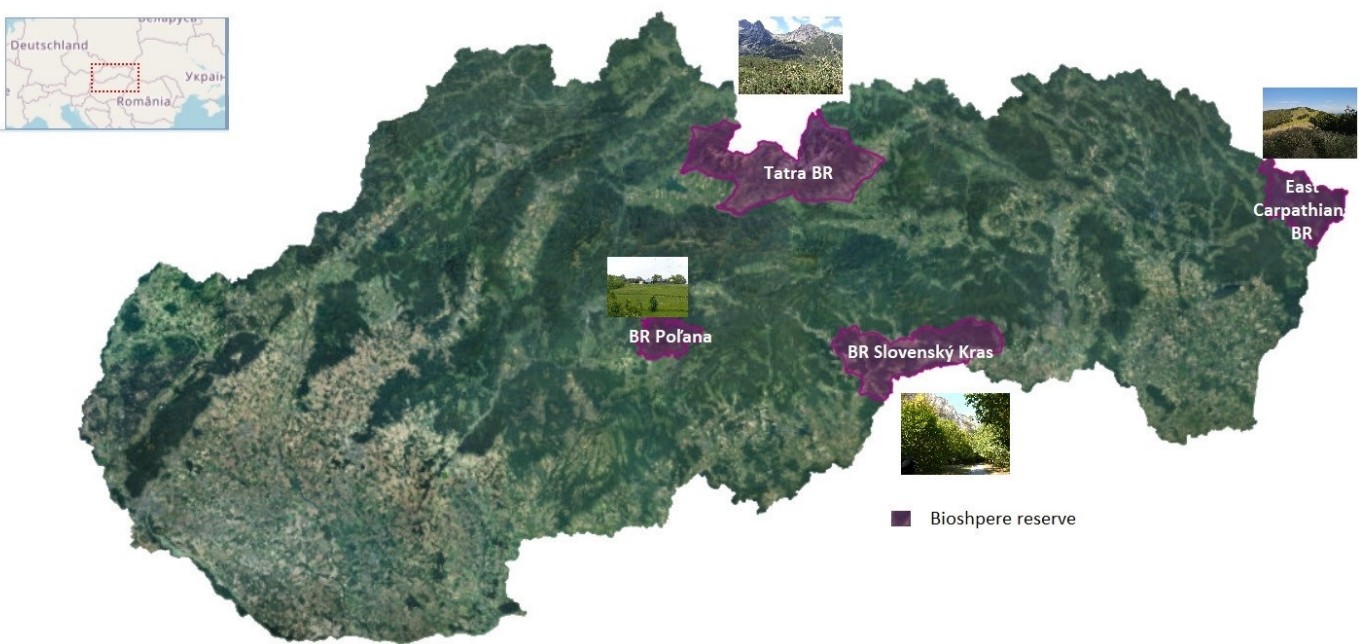

**Figure 1.** Map of the biosphere reserve in Slovakia [7,8].

**Table 1.** The main characteristics of the four biospheres in Slovakia.

| | Slovenský Kras Biosphere Reserve | Poľana | Tatra Transboundary Biosphere Reserve | East Carpathians Transboundary Biosphere Reserve |
|---|---|---|---|---|
| Year of designation | 1977 | 1990 | 1992 | 1992 |
| National protected area | NP (since 2002, from former PLA) IBA Slovenský Kras | PLA IBA Poľana | NP IBA Tatry | NP IBA Bukovské vrchy |
| Area | 74,500 ha (Core zone—8857 ha; Buffer zone—23,395; Transition zone: 42,248) | 24,158 ha—proposal for extension in 2015 (Core zone—1333 ha; Buffer zone—7930 ha; Transition zone—11,097 ha/proposal for extension 16,407 ha) | 134,448 ha, (Core zone—57,211 ha (Poland: 7548 ha—Slovakia: 49,663 ha), Buffer zone of 30,115 ha (Poland: 6371 ha—Slovakia: 23,744 ha); Transition zone—47,122 ha | 208,076 ha (Core zone—30,142 ha; Buffer zone—24,757 ha; the Transition zone—153,177 ha) |
| Development function—main activities | tourism associated with visiting caves; agriculture; mining and processing of raw materials, engineering, and metal industry | agriculture, forestry, tourism, and recreation | tourism; forestry | forestry |
| Socio-economic characteristics | about 47,900 people live in the transition area | almost 22,000 inhabitants have some involvement in the Poľana BR, 3900 of them live permanently in the transition zone | more than 40,000 inhabitants, of which a large majority live on the Slovak side. | 2299 inhabitants (in 2017, a declining trend) |

Legend: PLA—Protected Landscape Area, NP—National park, IBA—Important Birds Area.

The karst area is the richest district of the Pannonian flora, including xerothermous species, calcareous species, mountain dealpine and prealpine species in inversion locations, and important endemic, sub-endemic, and relict species. They create habitats for zoocenoses of the steppe and forest-steppe zone, endemic species of cave spaces, and xerotherm habitats

for butterflies, termites, beetles, and reptiles. Forest and smaller areas by steppe and forest-steppe zones cover most of the area.

### 2.1.2. Poľana Biosphere Reserve

The Poľana BR is situated in central Slovakia within the Western Carpathians mountain range. Poľana is one of the largest European former volcanoes (c. 2500 m above sea level) and is the highest volcanic mountain in Slovakia. The relief ranges from 460 m to 1458 m above sea level.

Most of the region is covered by woodland (85% of the Polana BR), the rest being agricultural land, including grassland and pastures, except for 50 ha of water reservoir. The forest habitats are very diverse, from oak forests to spruce forests growing on andesites, well known for their southernmost occurrence within the Western Carpathians, covering the highest part of the mountain range. Many forests have the character of an old primeval forest.

Poľana BR is characterized by frequent co-occurrence of both thermophilous and mountain plant species. There are about 1220 species of higher plants in the area, and it is also rich in lichens and mosses, including about 80 protected, threatened, or rare species. The variety and species-richness of the fauna in the Poľana BR reflect its environmental diversity, including significant protected, endemic, or endangered animal species. The fauna of beetles, butterflies, reptiles, and birds as well as large predators including wolf (*Canis lupus*), Eurasian brown bear (*Ursus arctos arctos*), and lynx (*Lynx lynx*) are particularly species-rich. The avifauna is exceptionally rich: there are 174 recorded species of birds in the area, including endangered and rare bird species.

### 2.1.3. Tatra Transboundary Biosphere Reserve, Poland/Slovakia

The Tatra Mountains are the highest mountains of the long Carpathian arc, which stretches from Slovakia to Romania via Hungary, Poland, and Ukraine. The territory of the BR covers two national parks on each side of the border between Poland and Slovakia. The Tatras cover five climatic zones and contain plants from lower montane forests, subalpine forests, dwarf mountain pine, alpine grasslands, and sub-nival zone. A variety of natural features are represented within this reserve, such as karst topography with dolomites and limestone, canyons, and waterfalls, a dwarf pine belt, alpine meadows, lakes, and rocky peaks.

Among the nearly 1300 species of plants in the Tatras, the most valuable are the 27 endemic and sub-endemic species. There are also numerous relics of the Pliocene and glacial flora. Numerous animal species are found in the BR, including the largest European predators: the European brown bear (*Ursus arctos arctos*), the Eurasian lynx (*Lynx lynx*), and the European wolf (*Canis lupus lupus*). High-altitude species such as the marmot (*Marmota marmota*), the snow vole (*Chionomys nivalis*), and the chamois (*Rupicapra rupicapra*) can also be found. Since 2000, the chamois population has increased by almost 850 due to the efforts of the Tatra Chamois Rescue Project.

### 2.1.4. East Carpathians Transboundary Biosphere Reserve, Poland/Slovakia/Ukraine

The East Carpathians Biosphere Reserve is a transboundary mountain reserve located in Central Europe that encompasses areas of significant value for biodiversity conservation. It covers the western edge of the Eastern Carpathians stretching across Poland, Slovakia, and Ukraine. Primeval forest fragments in this area have been preserved up to the present day. Large portions of the territory comprise forest and non-forest ecosystems, accompanied by numerous plant communities growing in semi-natural rural areas. Four distinct vegetation types are found within the biosphere reserve: beech forest (*Fagetum sylvaticae*), beech-fir forest (*Fageto-Abietum*), dwarf-shrublands with green alder (*Alnetum viridis*), and a belt of treeless 'poloniny'—subalpine meadows dominated by Prata subalpine. The European mountainous areas of the reserve are among the few to possess well-preserved native flora and fauna. In particular, the mixed Carpathian forest provides suitable conditions

for large mammals such as the European bison (*Bison bonasus*), the European brown bear (*Ursus arctos arctos*), the Eurasian lynx (*Lynx lynx*), and the European wolf (*Canis lupus*). Over 100 species of birds live in the area, including the black stork (*Ciconia nigra*) and the golden eagle (*Aquila chrysaetos*). The reserve also contains several sites appearing on the UNESCO World Heritage List, including the wooden churches of the Carpathian Region in Poland, Slovakia, and Ukraine, and the Primeval Beech Forests of the Carpathians in Slovakia and Ukraine.

### 2.2. Methods

An analysis and overview of previous research, as summarized in the action plans or management plans of the BRs by the administrative offices of protected areas, showed a high diversity of research studies, reports, and contributions: 264 contributions for BR Slovak Karst; 771 contributions for BR Poľana; 115 contributions for BR East Carpathians; 1051 contribution for BR High Tatras and the numbers are growing each year. Many of these studies are not officially published or open-access published, therefore, we based our methodological approach on the literature review and the selection of studies found in the scientific database Web of Science through keyword searches.

The selection of the studies was conducted through a keyword search in the largest scientific database, the Web of Science, published between 1990 and 2022 [9]. Publications containing the terms "biosphere reserves Slovakia", or, "the names of individual biosphere reserves" in the title, abstract, or keywords were selected using Boolean operators. The selection process for publications was performed according to PRISMA (Preferred Reporting Items for Systematic Review) [10]. Our selection returned 121 studies, which were analyzed according to Web of Science Categories and other detailed characteristics. The studies examined in the literature review were characterized according to the study area (target BR; and whether the study area is directly related to BR, or is rather a protected area like PLA or NP that is part of the BR, and the BR is only mentioned), research subject, BR function examined, and which aspect of the study contributes to the development of (Table 2). We investigated the research subject regarding the extent to which it reflects the mission of the scientific program Man and Biosphere, which is supposed to combine research in the field of natural and social sciences or whether it is focused only on some of these components. Based on this, we defined 10 research topics addressed by the reviewed articles, from studies of natural conditions (including abiotic and biotic conditions), social science (with emphasis on human perception and participation, the relationship including issues of biodiversity conservation interlinked with man, or land use changes; to optimizing the protection of biodiversity and natural capital of the biosphere (management of BR, measures for sustainable development, scenario modeling) and contribution of sciences, education or training. So far we have shown how BR work appears to foster learning about different domains, and aspects came from the way that the BR concept is situated "in-between" ecological, social, and economic goals, and reflects society's needs for quality of environment—environmental aspect (EN); needs of nature conservation—Ecological aspect (EO), cost to society (economic alternatives)—Economic aspect (EC); and benefits for society—Social aspect (SO). Other questions by our article review were open questions that gave a scientist's perception of the development of BR in a SWOT analysis, including identification of the Strength (main message of article/added value), Weaknesses (conflicts of interest, problem identifications, and shortcomings of research or management), Opportunities and Threats (recommendations for further development of BR or other comments).

**Table 2.** Criteria for analyses of selected studies dedicated to biosphere reserve research in Slovakia.

| Research Subject | Studied Characteristic |
|---|---|
| Target biosphere reserve | BR Slovenský Kras;<br>BR Poľana,<br>Tatra Transboundary BR;<br>East Carpathians Transboundary BR;<br>Several Slovak BR;<br>Slovak BR in an international context (Slovak BR and foreign BR) |
| Study area | 1. A BR or part thereof directly (1A. Core area; 1B. Buffer zone; 1C. Transition zone; 1D. no mention of zones)<br>2. protected area (PLA or NP, only mention of BR) |
| Study of the relationship between Man and the Biosphere | 1. Yes, directly deals with BR issues,<br>2. Indirectly (other research topics implemented in BR, or related topics, e.g., changes in land use)<br>3. Only biodiversity issue |
| Research topic | • Abiotic conditions;<br>• Biodiversity protection (different studies of biodiversity itself; from single species to habitats);<br>• Relation of biodiversity conservation interlinked with man;<br>• Land use changes;<br>• Ecosystem services (provision of benefits of ecosystem services to human society);<br>• Potential or impact of tourism and recreation on the biosphere;<br>• Social sciences—survey, perception of society, participation, stakeholder involvement;<br>• Management of BR, sustainable development;<br>• Scenarios modeling;<br>• Sciences, education, or training. |
| Research aspects | EN—Environmental;<br>EO—Ecological;<br>EC—Economic;<br>SO—Social |
| Message/added value, strength | Open question |
| Weakness, conflicts of interest | Open question |
| Recommendation for management and sustainable development of BR | Open question |
| Comments | Open question |

## 3. Review of Research in Biosphere Reserves in Slovakia

### 3.1. Review of Scientific Studies

The majority of the 121 analyzed studies were classified according to the Web of Science Categories into the category of ecology (61) or related environmental and biological sciences (Figure 2). The areas on the chart are not strictly proportional to the values of each entry, because some journals are included in several WOS categories; some categories are similar or overlapping.

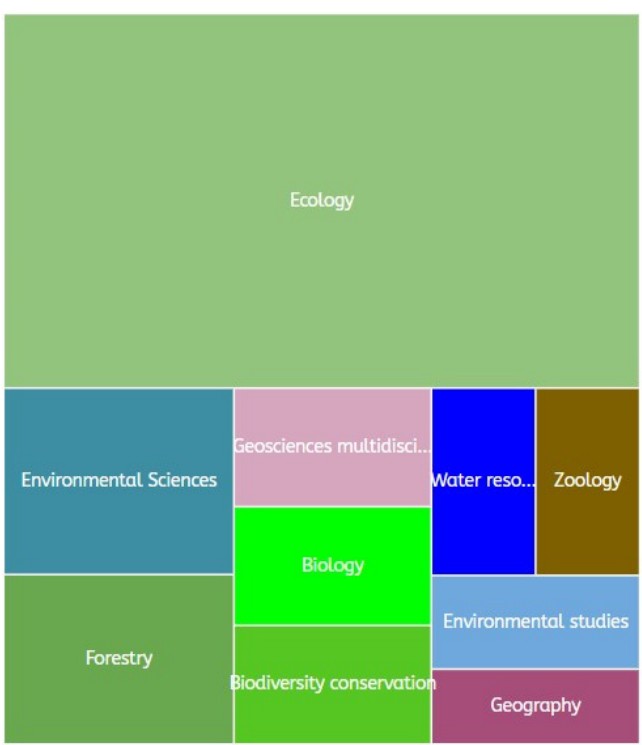

**Figure 2.** Visualization of publication search selected from all databases of Web of Sciences—TreeMAP chart.

Most of the papers are dedicated to specific research in one of the Slovak BRs; approximately 20% of papers are devoted to selected topics in multiple BRs, either at the national or international level (Figure 3). The highest number was published for BR Poľana, followed by Tatra Transboundary BR, the East Carpathians Transboundary BR, and the least for BR Slovenský kras. Our analysis shows that transboundary BRs in particular are involved in developing international collaborations within the World Network of BRs for societal transformation. The fact that scientists have displayed the most interest in Poľana BR is due to the proximity of and good cooperation with the Technical University of Zvolen (Faculty of Ecology and Environment, Forest Faculty) and other research institutes (Institute of Forestry of the Slovak Academy of Sciences and Forestry Research Institute in Zvolen).

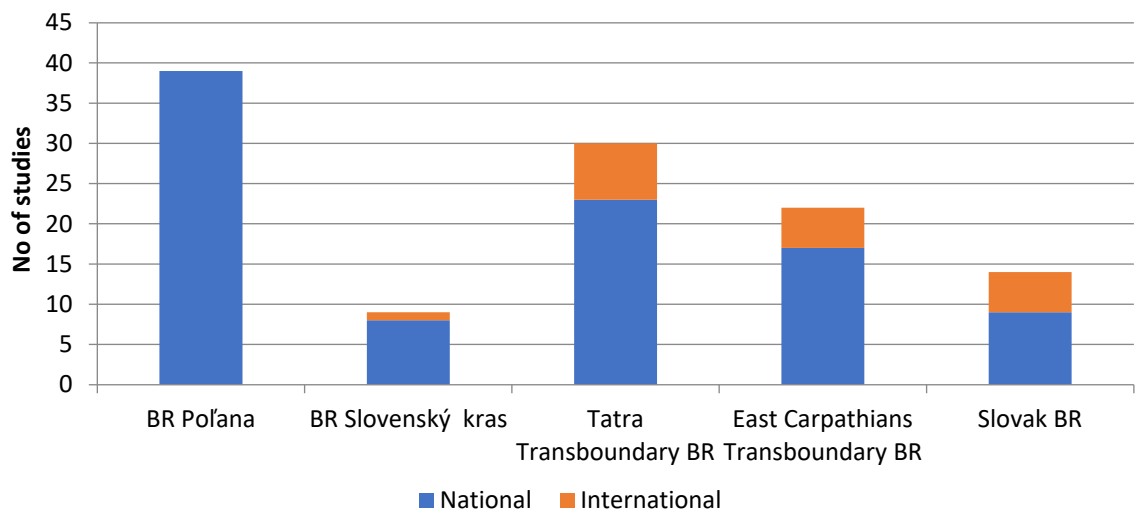

**Figure 3.** The overview of the target study area of Biosphere reserves in a national or international research context.

The study area is directly related to the BR for 65 studies (54%), while in 56 studies (46%) there is only a mention of the BR (Figure 4). Less than a third of the studies directly address the topic of the relationship between Man and the Biosphere, while others address it indirectly, and a significant part of studies are focused only on the issue of biodiversity. Most of the research was not related directly to BR zones; BR zones were mentioned only in two cases.

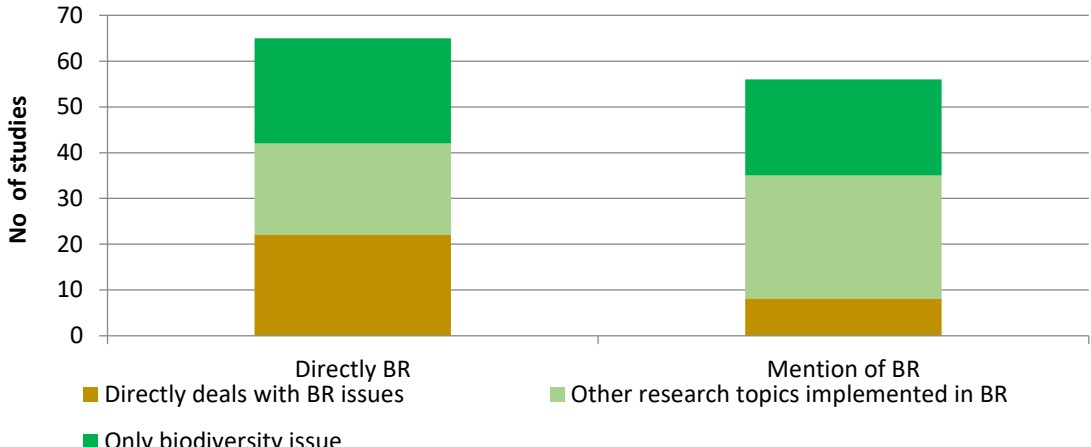

**Figure 4.** The issues of the relationship between Man and Biosphere in the study areas (BR studied directly as the study area, or only mentioned in the study).

A review of studies in BR revealed a wide range of research topics addressed in BRs (Figure 5), which contributes to the development of environmental, ecological, economic, or social aspects of BRs (Table 3). Most of the studies contribute to the ecological aspect, but there are also numerous studies combining different aspects of BR including the ecological, which constitutes an interdisciplinary approach to the important topic of man and the biosphere.

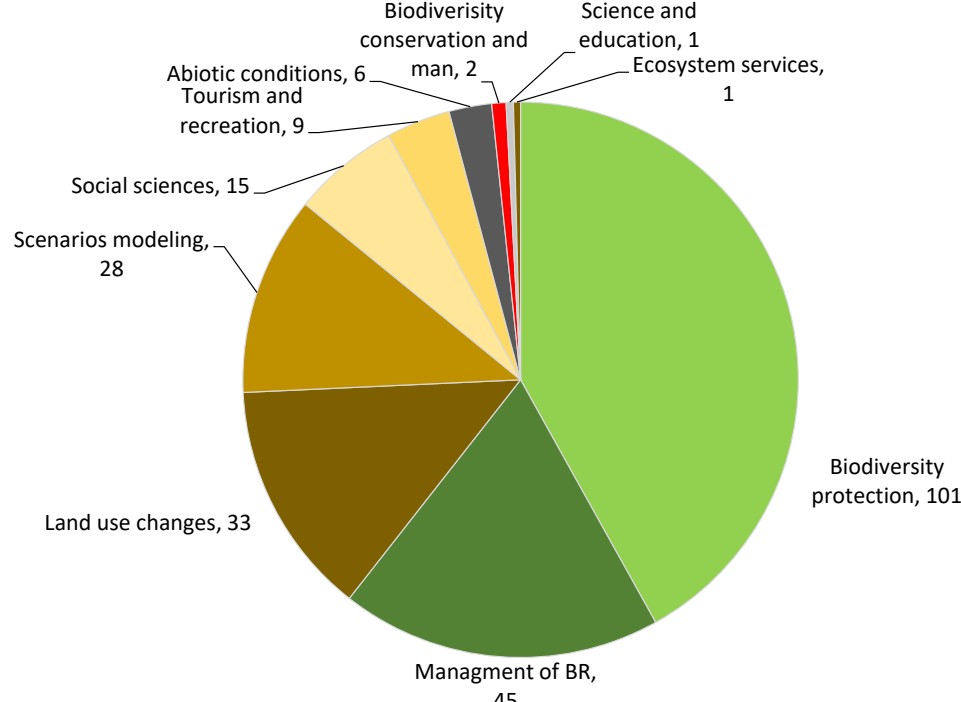

**Figure 5.** Research topic addressed in BR studies (in the number of studies).

**Table 3.** Researched biosphere reserves aspect in reviewed studies (EN—Environmental; EO—Ecological, EC—Economic; SO—Social).

| Studied Aspect | No of Studies |
|---|---|
| EN | 2 |
| EN, EO, SO | 22 |
| EN, EO, EC, SO | 1 |
| EN, SO | 7 |
| EO | 74 |
| EO, EC | 2 |
| EO, SO | 11 |
| SO | 1 |

*3.2. Specifics and Peculiarities of Individual BR*

The main research topics are devoted mostly to the specifics and peculiarities of individual BRs.

### 3.2.1. Peculiarities of Research in Slovenský Kras BR

The karst landscape of BR Slovenský Kras creates specific and unique abiotic conditions for the development of rare and vulnerable habitats, which is also the main subject of research in this area. Several studies focus on specific abiotic conditions and their threats in the context of land use changes and climate change, e. g. threats to karst lakes [11], assessment of soil organic carbon concentration and stock in the Silica Plateau [12], or the significance and value of the transboundary region of the Aggtelek National Park (Hungary) and the Slovak Karst Biosphere reserve (Slovakia), considering the geology, landscape geography, and cultural history of the region [13]. Specific natural conditions have made the area of Slovenský kras (Karst) to be a very rich district in terms of rare, endemic, or endangered species of flora and fauna [14–17].

The main threats to the biosphere of BR are biodiversity loss, degradation of habitats as a consequence of land use changes—related especially to the abandonment of non-forest habitats—and eroded abandoned lands caused by deforestation, burning, grazing, or improper land use and cultivation [18]. Climatic changes and the prevailingly negative impact of human activities, including agricultural intensification, cattle pasture, and artificial fishing, have a significant impact on the development of all lakes in the target area [11,13]. The main research findings included recommendations for biodiversity protection and sustainable management of threatened habitats, with special regard to sustainable silviculture or agroforestry [12,13].

### 3.2.2. Peculiarities of Research in BR Poľana

The review of scientific studies mainly deals with the topic of forest ecosystems and the landscape, which is reasonable due to the dominance of forests in BR Poľana. This research is devoted to different aspects of the forests and their environmental conditions: analysis of the vitality and health status of tree species, the ability of natural beech regeneration, the impact of acidity on spruce stands [19–21], the response of microbial activity on the temporal dynamics of soil water content in different forest habitats [22], contamination of water and soil by calamity wood exploitation, the eco-hydrological influence of Norway spruce and European beech on the hydro-physical properties of snow cover [23,24], water balance of the temperate forest ecosystem, the effect of forest management on biodiversity [25,26], and others.

A significant part of the studies focuses on land use changes in the cultural or agricultural landscape, pressures, and drivers, the impact of socio-economic changes (collectivization, transformation period after 1989), loss of the traditional agricultural landscape, increasing trend of abandonment and degradation of landscape function, assessment of flood risk and biodiversity threats [27–31]. The results of these studies highlighted the dynamic development of ecosystems of meadows and pastures over the last 70 years that,

at present, occupy approximately 10% of the territory, which has caused changes in the diversity of species, ecosystems, and landscapes [32–34]. Biodiversity research has resulted in records of many rare, endangered, vulnerable species of fauna and flora or even new species discovered in the PLA/BR Poľana [30,35–38].

### 3.2.3. Peculiarities of Research in Tatra Transboundary BR

The Tatra BR, with its diverse landscape and habitats, is an area of interest for various groups of scientists with a wide range of research topics: research on forests and non-forest habitats, biodiversity of fauna and flora, natural hazards, optimal land use, sustainable development, and sustainable tourism. Forest research is focused on various aspects of the environment and their impact on the vitality, health condition, structure, and dynamics of natural forests [39–42], biomass production, consequences of natural disasters (wind calamity or bark beetle calamity) [43], and responses of natural and managed forests to air pollution and other stresses [44]. Permanent monitoring sites have been established, studying fundamental abiotic and biotic conditions [45]. Due to the high natural values of the Tatra Mts, a variety of research, including on plants, animals, and mycological and zooplanktonic organisms, has been intensively conducted in this area for many years [46–48]. On-going monitoring is oriented towards the conservation status of numerous species and habitats of conservation importance, as well as the forest health status and other characteristics or indicators of forests. Abiotic conditions were studied in relation to natural hazards or the influence of local climatic and topographic conditions on the formation and development of the ice cover in high-mountain lakes [49–51].

The diverse values of BRs need to be understood and assessed as a basis for improving management effectiveness, as human activities cause various landscape-ecological and socio-economic problems. Several studies focused on setting up regulations (limits) for social and economic development with regard to landscape stability and nature protection, biodiversity conservation, rational use of natural resources and environment protection, as well as participatory management [52,53]. Tourism is an important link between conservation and sustainable development, as Tatra Mts is one of the most visited national parks in Slovakia. For this reason assessment of tourism's intensity and impact on nature is a useful tool for the selection of appropriate management measures to avoid or minimize negative consequences [54–56].

### 3.2.4. Peculiarities of Research in East Carpathians Transboundary BR

A large area of forest, including remnants of primeval forest, justified the establishment of a permanent study site for studying the effects of forest health on biodiversity, with an emphasis on air pollution in the Carpathian Mountains, where the species composition of particular forest stands and their geographical, ecological and floristic diversity, as well as their health, are monitored [57–60]. Even though forests are dominant in this area, most research is devoted to land use changes [61] and traditional agricultural landscapes with preserved valuable grassland [62,63]. In addition to species-rich grassland, interesting findings were also published for lichens, mollusks, and butterflies [64–66]. The area of East Carpathian was characterized by a significant change in landscape organization caused by the construction of a drinking water reservoir, which led to the emigration of local inhabitants (7 villages), abandonment of less accessible grassland, and changes in management due to the effects of the common agricultural policy. Studying land use changes led to identifying the driving forces of agrobiodiversity change and the implications for habitats and species, and predicting possible future trends in the region [67–69]. There is a need to support the maintenance of important landscape qualities through continued, low-intensive, farming methods, and more specific measures targeting especially valuable or important biodiversity and cultural heritage aspects, an approach which constitutes optimal land use from the perspective of the purposes of BR [70]. Nevertheless, ecotourism was found to be an opportunity in stimulating multi-functional and sustainable landscape management [71].

The BR serves as a Long-Term Socio-Ecological Research (LTSER) platform, which requires place-based transdisciplinary research infrastructure to attain the LTSER goal of a "learning landscape approach through evaluation" [72]. "Landscape approach" entails a collaborative effort of researchers, stakeholders, practitioners, and policymakers towards bottom-up projects and actions to promote a sustainable development process and sustainability in their own place and region.

### 3.3. SWOT

Based on the literature review and knowledge from other projects undertaken in BRs, we have summarized the perception of scientists on the development of BRs in the form of a SWOT analysis (Table 4). Based on that, we composed recommendations for logistic support, underpinning development through research, monitoring, education, and training in BRs.

**Table 4.** SWOT analysis of research in the biosphere reserve of Slovakia.

| Strengths | Weaknesses |
|---|---|
| ○ A long history and continuity of research, close cooperation with some scientists and institutions. | |
| ○ Existing permanent monitoring network of species and habitats of conservation importance on the national level (coordinated by State Nature Conservancy, reporting under Article 17 of the Habitats Directive). | |
| ○ Interest in research in the BRs by local, national, or foreign research institutes and universities—Slovenský kras BR (Comenius University Bratislava, Czech Academy of Sciences, Matej Bel University in Banská Bystrica, Pavol Jozef Šafárik University in Košice, Slovak Academy of Sciences), Poľana BR (Technical University in Zvolen, Comenius University Bratislava, Matej Bel University in Banská Bystrica, National Forest Centre, Slovak Academy of Sciences, Slovak University of Technology in Bratislava), Tatry Transboundary BR (Comenius University Bratislava, Czech Academy of Sciences, Matej Bel University in Banská Bystrica, National Forest Centre, Polish Academy of Sciences, Slovak Academy of Sciences, Slovak University of Technology in Bratislava, Technical University in Zvolen), East Carpathians Transboundary BR (Comenius University Bratislava, Pavol Jozef Šafárik University in Košice, Slovak Academy of Sciences). | ○ Lack of coordination of research in BR. |
| | ○ A significant fraction of studies is focused only on the issue of biodiversity. |
| | ○ Weak cooperation between the administration office of protected areas, research institutions, and university (mismatch of BR demand and supply of science, insufficient transmission of results). |
| | ○ Research activity depends on financial sources with related issues—non-continuity of research, the weak interest of scientists in research in some regions (e.g., Slovenský kras), or, contrarily, excessive numbers of scientists in other regions (e.g., Tatra BR). |
| ○ Wide range of research topics in Tatra Transboundary BR (conducted by a wide range of research institutes). | ○ Insufficient dissemination of the results to professionals, experts, and local inhabitants, would contribute to a better perception of the values and processes of the natural environment. |
| ○ Interdisciplinary research relating to Man and Biosphere (especially in East Carpathian Transboundary BR, Tatra Transboundary BR) | ○ Time-consuming processing of permits for scientists to enter the field and conduct research in protected areas of BR. |
| ○ Long-term monitoring sites (Poľana BR, Tatra Transboundary BR, East Carpathian Transboundary BR) as part of an ELTER (European Long Term Ecological Research) network implemented by research institutions. | |
| ○ Case study of BR included in international projects. | |
| ○ Existing examples of participatory studies. | |

**Table 4.** *Cont.*

| Opportunities | Threats |
|---|---|
| ○ The possibility of applying scientific knowledge, knowledge, and personal capacities necessary for developing the principles of sustainable development at the regional level (in action plans of activities for BRs). | |
| ○ Exchange of knowledge within the World Network of BRs, strengthening and expanding research questions in relation to Man and Biosphere | |
| ○ Demand for optimization of land use and landscape planning—extension and improvement of research relating to developing activities (e.g., tourism development and optimization of land use in Tatra BR is urgently needed, as is halting abandonment and development of the potential of tourism in other BRs). | ○ Insufficient human resources to develop collaborative research. <br> ○ Insufficient human resources for the education of natural-science-oriented university students. <br> ○ Limited financial sources related to the topic of BR research. |
| ○ Funding opportunity for transboundary cooperation through INTERREG Europe | ○ Changes of local governments (every 4 years, or more often) affecting the research funding in the next period |
| ○ Involvement of scientists' representatives in BR coordination boards (development of effective external partnership and specification of research strategy, definition of research activities in relation to demand versus supply) | |
| ○ Linking research with education for sustainable development and capacity building—outdoor and environmental education programs, and integration of the BR's heritage and culture within local education systems. | |

## 4. Discussion

BRs are 'learning places for sustainable development'. This statement was partially confirmed by our analysis in this review. The Lime Action Plan places a strong emphasis on thriving societies in harmony with the biosphere for the achievement of the Sustainable Development Goals and implementation of the 2030 Agenda for Sustainable Development, both within BR and beyond, through the global dissemination of the models of sustainability developed in BRs. Promoting sustainable development is a key difference between BRs and other forms of protected areas. In addition to the dominant ecological aspect of the research studies, the environmental and social aspects were also frequently considered, with social aspects discussed in approximately 12% of the reviewed articles. The applied social and economic aspect of research in BR is one of the opportunities for further development of BR that has to be strengthened. Lessons from two Canadian BR indicate that adaptive and reflexive community-based approaches offer methodological alternatives for research, help advance conceptions of community capacity, and help produce social change [73].

A significant part of the studies was focused on the protection of biodiversity, and the review findings highlight the exceptional values of the individual BRs in this regard, which need to be protected and maintained, especially in the core zone. Monitoring and assessment of biodiversity loss are one of the five global priorities identified by Evans [74], and one which requires the determination of changes in land use or land cover. The various values of the BRs need to be understood and assessed as a basis for the exercise of improving management effectiveness [4], as well as ensuring the long-term conservation of the socio-ecological systems of BRs, including restoration and appropriate management of degraded ecosystems. The results and conclusions of several revised interdisciplinary studies have pointed to changes in land use as a result of human activities. They help to understand and manage changes and interactions between social and ecological systems by developing proposals of measures and recommendations, including conflict prevention and biodiversity management. Understanding the causal relationships of individual land use changes, primarily related to agricultural activities, helped to model various scenarios and

develop a strategy for the sustainable development of the Poloniny National Park region of the East Carpathian BR [67]. The goal of such modeling is not for the local management in the landscape to choose one of the scenarios, but to take a realistic look at all of them, and to select issues to address and actions to take to benefit all participants while maintaining landscape conservation and biodiversity.

The analysis of scientific publications shows that most recommendations for management and sustainable development of BR strengthen the value of natural habitats, and emphasize the need for their conservation or improvement of conservation status, as well as support for agriculture and non-forest habitats, farming activities which provide local livelihoods as well as positive effects on natural values and biodiversity, possible forms of agritourism, attracting visitors to the region, and the overall economic growth of the region. The exception is the Tatras, which face the opposite pressure from investors and the need to regulate the carrying capacity of recreation in some of the areas most burdened by heavy tourist activities. One of the opportunities identified by the SWOT analysis is the harmonization of land use based on spatial differentiation and compliance with the principles of sustainable ecotourism. Another opportunity opens up here, to take advantage of outdoor and environmental education programs and the integration of BR heritage and culture into local education systems. Local communities benefit minimally from the development of tourism in BRs, although it takes place in their local territory, as living ecosystems are disturbed, and cars bring noise and pollution. In general, it can be stated that tourism in BRs should bring benefits to the environment of the local population and also to the tourism sector. Experience from foreign BRs shows that visitors can be interested in practical conservation activities during their vacation. Local tourism operators often turn to their clients with proposals to support the implementation of special local conservation projects. The fundamental issue is finding a sustainable balance between nature conservation and tourism.

However, tourism in BR is also related to agriculture. To create self-sufficient farms, farmers and other land users, such as tourists, must first build a collective awareness of this new role. In the long term, the farms should become self-sufficient. Activities in direct marketing can contribute to the achievement of this goal, e.g., the cultivation of old varieties of crops or using farms for agritourism. In the future, it will be important for farmers to use the whole farm, their products, and their traditional knowledge [75]. The review of scientific articles shows that the capacity of the Slovak BRs offers increased opportunities for the livelihoods and business of the local community and the ability to create a "good business climate".

With regards to pursuing their logistical functions, Slovak BRs do not have scientific teams, but many of them cooperate with various institutions in the field of research, education, training, communication, etc. Monitoring also plays an important role in logistical support. Analysis of research activities in BR revealed that various surveys of biota have been undertaken in the past and regular monitoring of habitats and species is ongoing, but the results of them are stored in databases or saved as survey reports, but not officially published. This highlights the differences in needs and goals of research implementation between research institutions and administration/management bodies of protected areas. While research institutions are focused on studying processes and definitions of project tasks with output in scientific articles, management bodies of protected areas regard knowledge about the subject of nature conservation as beneficial and apply it to management plans. Acquiring knowledge through this process is the basis for assessing the initial state of the territory, and for appropriate management decision-making. At the same time, the mutual connection of these "two worlds" (science and research, and application practice in the form of suitable management measures) is important, which can be beneficial for both partners and the sustainable development of the whole BR. A research strategy needs to be developed based on the gaps identified in the literature. A big challenge is to find a common language and build bridges between the demand from practice and the supply of science. The positive point is that permanent monitoring

sites have been established in three out of four BRs in Slovakia, where the development and changes in biodiversity are monitored and studied. Data sharing increases the scope of the biosphere reserve's impact. Additionally, lessons learned from local solutions may assist with facing those global challenges which the World Network of Biosphere Reserves was established to address. Currently, coordination councils of BRs are in the process of being updated, and contractual cooperation is being created between scientific institutions, universities, and BR administrations. Our study can help guide these teams, as it indicates the particularities of research in individual BRs. It is important to set up closer cooperation between scientists and those engaged in practical conservation management, which the Administration of the BR would indicate scientific areas and topics of particular interest, and scientific teams would work on assigned tasks.

According to our SWOT analyses, the key to reducing the manifestation of undesirable pressures in the environment is the strengthening of environmental education and training and the improvement of environmental policy so that even with limited financial resources there is the political will to direct these resources to the protection of the environment. Every BR is obliged to develop and implement criteria and contents for environmental education in its framework plan, considering the specific structures of the BR. There should be diverse environmental education offers for the population and visitors in BRs: natural history excursions and seminars, nature experience programs, project days, teaching and nature experience paths as well as information centers with exhibitions and extensive information offers. Various stakeholder groups should be able to obtain comprehensive information about the natural resources of the area, the objectives and tasks of BRs, and the relationship between people and the environment. Educational programs are popular abroad and establish BRs as important brands with increased external influence [76,77]. Moreover, it is important to ensure there are environmental education programs focused on BR in primary and secondary schools (especially in BR regions). In addition to education, the SWOT analysis also points out the importance of the inclusion of local communities in the creation and implementation of management plans and regulations [78], which is also not possible without education.

One important lesson to be learned from the experience of BRs in other countries is the value of a Communication and Educational Action Plan which includes the promotion of the values of BR and environmental education—as has been done in Spain and Costa Rica in particular, but also Malaysia, e.g., BR Tasich Chini [79]. This action plan should outline pedagogical activities that include teaching and guiding students to support their learning through continuous professional training. In this sense, the learning action plan is a part of the larger educational project of the BR and must be consistent with it. Education for sustainable development must be given a completely new standing within the functions of BRs: education and life-long learning are fundamental components of sustainable development, a process in which the global guiding principle of "sustainability" is constantly given practical definition by, and realized in the context of, new local and regional (sub)objectives. An educational and learning process of this kind must be supported by integrated research approaches in which not only natural and social sciences interact (ideally in an interdisciplinary way), but also the various groups of stakeholders in a BR are continuously included to participate. Research activities may include the options of Bachelor's and Doctoral Theses in BR, which can create space for mutually beneficial cooperation.

Communication and cooperation are essential for the control of activities in BRs and conflict solutions. The local and regional stakeholders, including active individuals with knowledge and experience, are the pool of experts for their region [80,81]. They are the most important communication and cooperation partners of every BR administration.

Within the World Network of Biosphere Reserves, the number of transboundary BRs is rising. These are BRs that are on both (or all) sides of a political border, and the countries involved officially announce their intention to cooperate in the protection and sustainable use of an ecosystem by means of joint management [82]. These BRs provide numerous advantages, such as successful conservation and sustainable use management, promotion of

regional development in peripheral areas, and preservation of cultural identity and integrity, as well as successful mitigation of conflicts, promotion of peace, and crisis prevention [83]. Only a few studies have dealt with transboundary research in the cross-border territories of the Tatras and the Slovak Karst.

As follows from the definition, goals, and function of BRs, they have to be engaged with international, regional, national, and subnational research initiatives and programs that should identify, understand and address present and future economic, environmental, ethical, and societal challenges related to sustainable development; research in Slovak BR both supports and participates in these initiatives and programs. Studies in various fields of natural, social, and human-environmental research highlight that the World Network serves as a forum for the co-production of knowledge for sustainable development [84]. Our review showed that BR Poľana primarily meets the defined BR criteria for the implementation of the MAB program in Slovakia. However, from the spectrum of publications, it is possible to see the need to connect the individual needs of other BRs. With the current creation of permanent scientific teams in individual BRs, new opportunities arise that enable the current BRs' vision.

## 5. Conclusions

BRs are sites for testing interdisciplinary approaches to understanding and managing changes and interactions between social and ecological systems, including conflict prevention and management of biodiversity. They are places that provide local solutions to global challenges. Each BR has its natural specificities and cultural and biological diversity that must be respected, and each BR offers unique local natural resources. Therefore, it is necessary to link research with education for sustainable development. Our review of 30 years of research in the Slovak BR helped us to highlight outputs, outcomes, impacts, and contributions to their management and sustainable development, and to draw conclusions regarding future challenges and developments for the research strategy of Slovak BR.

Slovak BRs have a long history of research, continuing to the present day. The strength of these research activities is the numerous existing partnerships between BRs and universities/research institutions to undertake research. The many studies and monitoring programs carried out on the territory of BRs create the possibility of comparing current results and evaluating development and changes in the context of other socio-economic activities or creating predictive models associated with climate change. Most of the projects and published papers focused on biodiversity protection, management of BR, land use changes, and scenario modeling. Our analysis of 121 scientific studies shows that transboundary BRs in particular are involved in developing international collaborations within the World Network of BRs. A special contribution to research in Slovak BR is that provided by existing long-term monitoring sites, which have been included in international projects as case studies. It is necessary to ensure and maintain adequate research infrastructure in each BR and develop strategies for further research in which the supply of scientific research meets the practical requirements of the BR.

Based on our SWOT analysis we have composed recommendations for logistical support, and underpinning development through research, monitoring, education, and training in BRs. Among the strengths of BR in Slovakia are its long history and continuity of research, and close cooperation with some scientists and institutions. Strengths also include the existing permanent monitoring network of species and habitats of conservation importance on the national level (coordinated by the State Nature Conservancy, reporting under Article 17 of the Habitats Directive) and ELTER long-term monitoring sites; interest in research in the BRs by local, national or foreign research institutes and universities; and case studies of BR included in international projects, existing examples of participatory studies, and a wide range of research topics in the highest part of the Carpathians, the Tatra Transboundary BR.

The research also pointed to various weaknesses, such as insufficient coordination of research in BR; weak cooperation between the administration of protected areas, research

institutions, and universities; unstable financial resources for research; and insufficient dissemination of results to professionals, experts, and residents, which would contribute to a better perception of the values and processes of the natural environment.

Slovakia needs to strengthen the possibilities of research, practical education, and training that support BR management and sustainable development in BR (point A4 in the Lima action plan). It is also important to establish partnerships with educational and training institutions, and to undertake education, training, and capacity-building activities aimed at BR stakeholders, including managers and owners, taking into account the SDGs. To improve cooperation, it would be beneficial to involve scientists' representatives in the coordination councils of BR.

**Author Contributions:** Conceptualization and methodology, J.Š.; formal analysis, resources, N.M.; writing—review and editing, J.Š. and V.P.; visualization, N.M and J.Š.; project administration and funding acquisition, V.P. All authors have read and agreed to the published version of the manuscript.

**Funding:** This research was supported by Project APVV-20-0108 Implementation of Agenda 2030 through biosphere reserves.

**Institutional Review Board Statement:** Not applicable.

**Informed Consent Statement:** Not applicable.

**Data Availability Statement:** Not applicable.

**Conflicts of Interest:** The authors declare no conflict of interest.

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
