# Peer review of "The Contribution of Scientists to the Research in Biosphere Reserves in Slovakia"

_land, doi:10.3390/land12030537_

Round 1
Reviewer 1 Report
The study aims at summarising the current state of knowledge of BRs in Slovakia which is particularly important field.
In the Introduction the authors mention that: “The gap between the designation objectives of BRs and their management is often very wide [3]. Therefore, the BRs’ management needs to be reinforced by focused research and by application of research results for sustainable development and support for mitigation of and adaptation to climate change and other global 55 environmental changes”. Which is totally true, but I suggest tofurther explain the what kind of good practices from published research should be used in the management of BR.
In the methodology the Criteria for analyses of selected studies dedicated to biosphere reserve research in Slovakia is mentioned only in the table, but the performed analysis is described in a very general way. The swot analysis is presented only in the results section, and it is not even mentioned in the methodology. Also, the authors should clearly mention the type of research – is this a review paper / a bibliometric paper etc. and clearly make the difference.
The color palette of the figures is the automatic one, I advise the authors to use something customized; the quality of the figures should be high as required in a scientific article.
Discussion must make direct reference to the research findings and also they should be compared to other studies related.
Author Response
Dear reviewer
thank you very much for your effort and detailed comments.
We give a point-by-point reply to your comments in attached file.
We have revised the whole manuscript based on your recommendations and we hope that this increases the readability and scientific quality of the manuscript.

Reviewer 2 Report
The article is devoted to an important and actual topic of the connection between scientific research and the management of protected areas, in this case, these are four biosphere reserves in Slovakia. However, the article is replete with both fundamental semantic shortcomings and smaller technical and linguistic errors. In my opinion, the article can be useful to readers only after they have been corrected by the authors.
1. METHODS. The criteria chosen by the authors for the analysis of articles (Table 2) are partially not described, partially illogical, partially incomprehensible. The method for obtaining the results shown in Figs. 2 not described.
2. RESULTS.
- The results are shown in different units - either the number of articles, or their share as a percentage, or incomprehensible percentages corresponding to parts of the articles (Fig. 5). Units of measurement are not labeled on the figures.
- The share of M&B studies was counted as 1/3 in Fig.4 and as 0.8% in Fig. 5.
- The results of SWOT analysis do not correspond to the logic of this method. Is it an analysis of the strategic planning of general management in BRs or only scientific research in BRs? If this is an analysis of BR management, then many important issues are missing related to protection, visitor regulation, territorial planning, etc. If this analysis concerns only research, then there are extra points that do not directly affect research (e.g., bullets 4 and 5 from Threats do not directly threaten research). The block of “Opportunities” contains a list of things to do, but not an analysis of opportunities that should be favorable external factors that the analyzed system (BR+researchers) does not control. Some of the items in Table 4 repeat or contradict each other.
3. DISCUSSION. This section contains general reasoning is presented, but no discussion of the results presented in Section 3.1 is provided.
4. CONCLUSIONS. The conclusions are not related to the results of the literature analysis obtained by the authors (Section 3.1). The task stated by the authors in the Introduction (lines 75-76) to show how publications contribute to the mission and fulfillments of BR’s goals and building the World Network of Biosphere Reserves has not been fulfilled
Detailed comments (also see attached PDF file)
Lines 10-23. In the abstract, it is desirable to summarize the results of the study. I didn't find them here.
Line 12. Mentioning "three functions" in the abstract is confusing. It is advisable to either remove the "three functions", or explain what they are
Lines 75-76. It's left unexplained
Lines 169-170. Is this your research data? Need to explain
Lines 174-175. The meaning of these aspects is unclear. See comment below for Table 2
Line 182. How were articles divided among these sciences? see comment below for fig. 2. And this refers to results rather than methods
Line 183. Table 2.
2nd row. What is the protected area in this case?
3rd row. What other topics? It follows from the following text that these may be topics not related to M&B relationships, for example, abiotic conditions. “only biodiversity issue” – is it means NO M&B studies? If this row has both YES and NO answers, then it is better to change its name (presence of M&B studies?)
4th row. It's a poorly structured mix of topics:
1) and 3) - what is the research difference between "sustainable development" and "M&B relationships"?
2) - studies of biodiversity itself, or measures for its protection, which refers more to point 1?
6) - scenario modeling can include all the topics mentioned here
7) and 8) - tourism and recreation are ecosystem services
5th row. The names of these aspects are incomprehensible and confusing. It may be necessary to change their names, as the explanations in brackets are more understandable. Can't EN and SO be expressed in economic terms? Can't EO be environmental? Can't EN be ecological? "Cost of society" - do you mean "cost FOR society"?
6th – 9th rows. Why these points, if you have not written anything about them?
Line 185. Figure 2. These are results rather than methods. But there is no description in the methods of how these results were obtained. Based on what criteria were articles assigned to these topics? Table 2 lists other topics. What is the difference between "Environmental studies" and "Environmental sciences"? And "water resources" belong to this topic. What is the difference between "Geography" and "Geosciences multidisciplinary " if Geography is multidisciplinary science? Zoology is a part of Biology. And most of the topics shown here belong to Ecology. That is, here Ecology is not Biology, is not Environmental science. So, what is it?
Line 188. Unfortunate subtitle, as this is the general theme of the work
Figure 3. Is it a number or a percentage?
Figure 4. Is it a number or a percentage?
Line 212. Is it 54% from 121, i.e. 65 papers?
Line 216. This approach is not ecological? See above comment about titles of "aspects"
Figure 5. Are parts of articles counted here? 0.1% from 121 is 0,1 article. 0.1% from 65 is 0,06 article.
Line 220-221. See above comment about titles of "aspects"
Line 222. Maybe it makes sense to title it like this: "specifics and peculiarities of individual BRs"?.
Line 320. Section 3.3. Is it an analysis of the strategic planning of general management in BRs or only scientific research in BRs? If this is an analysis of BR management, then many important issues are missing related to protection, visitor regulation, territorial planning, etc. If this analysis concerns only research, then there are extra points that do not directly affect research (e.g., bullets 4 and 5 from Threats do not directly threaten research).
Table 4
- Bullet 5 of Strengths contradicts bullet 2 of Weakness
- Bullet 6 of Strengths is repetition of the bullet 2
- Bullet 5 of Weaknesses is a part of bullet 3
- Bullets 4 and5 of Threats are threats to the BR itself, but not to research.
- Block “Opportunities” contain a list of things to do, not an analysis of opportunities.
- Bullet 3 of Opportunities - probably not just development, but optimization of tourism in the Tatras? More precise wording needed

Author Response

(The authors gave the same response as above.)

Round 2
Reviewer 1 Report
The manuscript is much improved and can be considered for publication in its current form.
Author Response
Dear reviewer
thank you very much for approval of publishing our manuscript.
Reviewer 2 Report
The authors corrected most of the shortcomings indicated in the review. For the future, I strongly advise authors to either highlight the changed pieces of text in the manuscript, or cite the changes made, indicating the lines in the corrected manuscript in response to reviewers. The qualification of the reviewer suggests that it is inappropriate to spend his time comparing two versions of the manuscript.
Despite the fact that most of the shortcomings have been corrected to one degree or another, a few small flaws remain, the improvement of which will be useful for the perception and understanding of the article by readers.
1) I still advise authors to more clearly reflect the results of the study in the abstract, namely, to briefly and clearly define exactly how research in the BR helps (does not help) solve their main task in the field of interaction between man and the biosphere. Believe me, I am aware that the number of words in the abstract is limited. There are enough general phrases in your abstract that you can remove from there without losing key information
2) The question about the relationship of research topics to the MAB concept (lines 194-200 of the second version of the manuscript and row 4 of Table 2) remained unresolved. Of the 10 topics shown in the table, all but the first two are, to one degree or another, related to the interaction of man and nature, that is, to the MAB program. If you explain this in the text and remove "MAB" from the third topic in the table, readers will not have this question. Your expert ranking of the 10 topics in terms of their relevance/belonging to the MAB concept would make your approach even clearer
3) lines 371-372 of the manuscript V2 ("social aspects discussed in approximately one third of 371 the reviewed articles") - it is not clear what data is being referred here. Fig. 5 shows that the social sciences are considered in only 15 out of 121 articles, that is, in 12% of the articles.
I think that after the authors have considered these minor shortcomings, my second review is no longer needed.
Author Response
Dear reviewer, thank you very much for your recommendation, which we accepted and we hope it improve the quality of manuscript. Response to your recommendation are in attadech file.
